# Home Environment Augmented Reality System Based on 3D Reconstruction of a Single Furniture Picture

**DOI:** 10.3390/s22114020

**Published:** 2022-05-26

**Authors:** Hongtao Wei, Lei Tang, Wenshuo Wang, Jiaming Zhang

**Affiliations:** College of Information Engineering, Wuhan University of Technology, Wuhan 430070, China; weiht@whut.edu.cn (H.W.); wws1049722105054@whut.edu.cn (W.W.)

**Keywords:** 3D reconstruction, instance segmentation, 3D registration, furniture objects

## Abstract

With the popularization of the concept of “metaverse”, Augmented Reality (AR) technology is slowly being applied to people’s daily life as its underlying technology support. In recent years, rapid 3D reconstruction of interior furniture to meet AR shopping needs has become a new method. In this paper, a virtual home environment system is designed and the related core technologies in the system are studied. Background removal and instance segmentation are performed for furniture images containing complex backgrounds, and a Bayesian Classifier and GrabCut (BCGC) algorithm is proposed to improve on the traditional foreground background separation technique. The reconstruction part takes the classical occupancy network reconstruction algorithm as the network basis and proposes a precise occupancy network (PONet) algorithm, which can reconstruct the structural details of furniture images, and the model accuracy is improved. Because the traditional 3D registration model is prone to the problems of model position shift and inaccurate matching with the scene, the AKAZE-based tracking registration algorithm is improved, and a Multiple Filtering-AKAZE (MF-AKAZE) based on AKAZE is proposed to remove the matching points. The matching accuracy is increased by improving the RANSAC filtering mis-matching algorithm based on further screening of the matching results. Finally, the system is verified to realize the function of the AR visualization furniture model, which can better complete the reconstruction as well as registration effect.

## 1. Introduction

With the impact of the global new crown epidemic, shopping without leaving home is encouraged by the state to reduce the spread of the epidemic brought about by the gathering of people. Online shopping for small items usually does not require consideration of size, and returns and exchanges are easier to carry, but online shopping for medium to large furniture products can have certain drawbacks. Traditional furniture procurement is mainly based on subjective feelings to infer the placement of furniture in the room and placement effect; this approach will cause the purchase of furniture back in the actual placement effect and the size of the house and furniture color with the incongruity. The return of furniture products is quite time consuming and laborious, so the current furniture procurement model is mostly offline procurement. Offline furniture city furniture format is relatively fixed and cannot provide users in the shopping site favorite personalized furniture products, according to the website furniture picture to get the same model for real-time home experience product suitability of the new way to call out.

Augmented Reality (AR) technology, as an important technical support for the underlying technology of “metaverse”, displays virtual models or information in the real world for interaction, which can usually assist users to display and work in reality [1]. With the increase in the computing power of electronic products, AR technology is used in all aspects of people’s lives. Viewing a single image of furniture on a shopping website and creating a 3D model that can be used for AR in real time has become a new way of shopping [2]. Although many scholars have investigated how to reconstruct 3D models from single images, the premise is based on clean, background-free example objects. Since furniture images on websites usually contain complex backgrounds, three problems need to be addressed: first, the need to extract clean furniture images from complex backgrounds; second, 3D reconstruction based on individual furniture images; and third, accurate registration of the reconstructed 3D models into the real environment.

This paper aims to combine 3D reconstruction technology and augmented reality related technology to develop an AR home environment system capable of real-time 3D reconstruction based on furniture pictures, as well as AR browsing and furniture placement based on users’ online shopping furniture product scenes. The system can basically satisfy the user to reconstruct the furniture pictures with complex backgrounds when browsing the shopping website and pre-process the pictures to remove the backgrounds to get a clean furniture example picture, reconstruct the 2D furniture pictures in 3D to get a 3D model of furniture, and then use AR technology to apply the furniture model to arrange the home environment to realize the virtual shopping function. The main contributions of this paper are the following.

This paper proposes a Bayesian image background segmentation (Bayesian Classifier and GrabCut, BCGC) algorithm, which can segment the furniture instance map more accurately by combining a Bayesian classifier with the GrabCut algorithm;Based on the advanced 3D reconstruction network ONet network, a precise occupancy network (PONet) is proposed. PONet adds camera information, combines local features as well as global feature information, adds a two-branch probability estimation, and finally outputs the occupancy probability of each of its points to achieve the effect of 3D mesh reconstruction model;Based on the markerless tracking registration method, the AKAZE algorithm is improved, and a Multiple Filtering-AKAZE (MF-AKAZE) algorithm is proposed to achieve real-time tracking and localization of the model.

## 2. Related Work

The reconstruction of 3D models from single images has become a hot topic in recent years with the widespread use of deep learning, and research on 3D reconstruction by deep learning has been accelerated by the proposal of large 3D model datasets and end-to-end learning frameworks [3]. Inspired by the ability to achieve reconstruction using a priori information, Choy [4] proposed a 3D recurrent neural network 3D-R2N2 (3D Recurrent Reconstruction Neural Network), which learns 2D images from an image dataset, establishes a mapping relationship between the 3D model and the image, and infers a voxelized model based on multiple images of the same object. 

Although the voxel representation method can reconstruct a 3D model of an object based on a single image, the reconstructed model is mostly a 3D model with low resolution due to ignoring the natural invariance of the 3D shape and therefore has certain drawbacks. The point cloud-based representation method scans the set of points on the outer surface of the object by the device, which can effectively overcome the drawback of the natural invariance of the object under geometric transformation using the voxel method. Hao [5] proposed a point set generation network (PSGN) based on point sets, and the network designed a bar shape sampler to predict multiple point clouds based on a single picture. The network is designed with a bar shape sampler to predict multiple point cloud forms based on a single image, solving the problem of a large number of unsuitable models. A hourglass convolutional network is introduced in the network structure to iteratively encode and decode it to produce better aggregation capability [6]. 

It is difficult to reconstruct a 3D model with a continuous surface to be applied to a real scene because the point cloud method loses information on the model surface. To solve the above problems, Kato [7] proposed a rasterized approximate gradient that integrates the rendering of the model into a neural network and performs single-image 3D reconstruction using contour image supervision. After the experiments, it was shown that this approach is more intuitive and better reconstruction compared to the point cloud-based representation. Kong [8] estimated dense 3D models from single images using a low-dimensional embedding space, which preserves the geometric features of the models.

In recent years, an implicit surface representation-based 3D reconstruction of objects has gradually been developed. Li et al. [9] proposed a Field Probing Neural Network (FPNN) to utilize 3D fields as the basic representation of 3D shapes. The algorithm is inspired by the inefficient voxel approach, and instead of 3D convolution computation uses multiple detection filters to sample the 3D fields. Park et al. [10] proposed a network structure called DeepSDF for learning the symbolic distance function of the shape representation. This learning framework represents the shape as a continuous 3D field, encodes the boundary of the shape as a zero level set of the learning function, and encodes the whole space with positive and negative symbolic distance lengths Distinguish whether the feature points belong to the exterior or interior. The advantage of this model is that it can represent not only the SDF of a single shape but also a class of shapes. Meschder et al. [11] proposed an Occupancy Network (ONet) architecture capable of reconstructing the space of 3D shape functions from different input patterns. The network framework formulates the reconstruction task as a binary classification problem, and the surface of the model represents the decision boundary of the classifier. The basic idea of the algorithm is to pre-train a ResNet image encoder and send a 2D image as input to the encoder to generate an embedding representation of the input. Peng et al. [12] proposed a convolutional occupancy network that combines a convolutional encoder with an implicit occupancy decoder for structured reasoning in 3D space, which can intern fine-grained implicit 3D reconstruction of a single object.

AR technology is characterized by a combination of virtual reality, real-time interaction, and three-dimensional (3D) alignment [13]. Among them, 3D alignment is the core of the AR system and a key metric to evaluate the performance of the AR system, which is to solve the real-time mapping between 3D target objects in the real world and two-dimensional (2D) screens [14]. However, the natural feature-based 3D alignment method takes some time to extract feature points and is prone to feature point mismatches, requiring a computational process to remove redundant points. This is a challenge to the real-time performance of the system, which cannot completely eliminate the mismatch [15]. Feature extraction and matching are key issues for natural feature-based alignment.

Currently, the more commonly used feature-based matching algorithms are Scale Invariant Feature Transformation (SIFT) [16], Accelerated Robust Feature (SURF) [17], and Oriented Rapid Rotation Sketch (ORB) [18]. Dai et al. [19] improved the real-time performance of tracking by combining SURF with ORB based on the dichotomous segmentation of images, but the robustness of the algorithm was degraded. A scale-invariant image matching method to address very large changes in the view was proposed by Zhou et al. [20]. However, the application in the built environment is vulnerable to factors such as illumination, scale, and angle, which leads to poor matching. Huang et al. [21] proposed a novel image matching method based on SIFT and LPP. Zhou et al. [22] proposed SIFT in GMS selected regions (SIGMA), an effective mismatch elimination based on SIFT and GMS point-based method. The above studies are all based on Gaussian kernel functions to construct a scale space. This will lead to the loss of image edge information, which will seriously affect the stability of feature points and descriptors [23]. For this reason, some scholars have started to study the nonlinear filtering function for constructing the scale space. This method can better preserve image details and has a stronger adaptability to scale changes. Currently, KAZE [24] and AKAZE [25] are mainstream algorithms. Experiments show that the above algorithms have stronger robustness than orb and SIFT.

## 3. Methods

### 3.1. System Architecture

The main function of this system is to realize 3D reconstruction and AR 3D registration of furniture pictures containing complex backgrounds, so it mainly includes three steps: the first part is to extract and segment clean furniture pictures from complex backgrounds; the second part is to get 3D models of furniture by 3D reconstruction of furniture pictures without complex backgrounds; and the third part is to make improvements to feature matching in 3D registration of models to achieve AR viewing and interaction. The main framework of the system is shown in Figure 1.

In order to realize the system to remove the background of furniture pictures containing complex backgrounds, this experiment proposes a BCGC algorithm in which the user draws a bounding rectangle to get the position of furniture instances in the image and extract them. In the reconstruction experiment, a PONet network is proposed to complete the probability calculation to realize 3D reconstruction based on pictures. In the 3D registration experiments, the traditional AKAZE feature detection algorithm is improved, and the MF-AKAZE algorithm is proposed for more accurate feature detection and matching to ensure that the 3D model of furniture can be registered to the real environment in a stable and accurate way to achieve AR interaction function.

### 3.2. BCGC Instance Segmentation

The Bayesian classifier (BC), as one of the most basic statistical classification methods, has an image segmentation extraction function. Image thresholding is a widely used segmentation technique that takes advantage of the difference in grayscale characteristics between the target region to be extracted from an image and its background, treats the image as a combination of two types of regions (target region and background region) with different grayscale levels, and selects a more reasonable threshold value to determine whether each pixel point in the image should belong to the target region or the background region to produce the corresponding binary image. The BC algorithm is able to segment the foreground and background where the instance is located, but the part of the hole contained inside the instance is not completely segmented, and further refinement of the segmentation is needed. GrabCut considers the global color distribution of background and foreground pixels (with local pixel similarity) for segmentation, so it has the ability to remove the internal pixels that do not belong to the object, and therefore uses the GrabCut algorithm once again segments the internal non-essential parts of the foreground with the foreground. The algorithm steps are shown in Algorithm 1.
**Algorithm 1** Step of BCGC**Input**: Image Iin, Bounding box, N: maximum number of iterations, Pixel Threshold T.**Output**: Image Iout.**Procedure**:  Step 1. Train an initial BC.  Step 2. BC algorithm distinguishes the foreground background according to GMM to get image IBC.  Step 3. For each pixel zi∈Z, the background similarity p(zi) is calculated by GMM.  Step 4. The image result IGC is further obtained by refining the segmentation with GC algorithm.  Step 5. if p(zi) ≤ T then Iout(zi)=IBC(zi) else Iout(zi)=IGC(zi).  Step 6. If the segmentation result is unchanged or the current iteration times is larger than N, the algorithm is over.  Step 7. Or else re-train BC based on the segmentation result, then go to Step 2.

The first step of the algorithm is to initialize the BC network. The posterior probability density function can be expressed as Equation (1).
(1)$$P\left(C_{i}|x\right)=\frac{p\left(x|C_{i}\right)P\left(C_{i}\right)}{\Sigma_{i=1}^{2}p\left(x|C_{i}\right)P\left(C_{i}\right)},i=1,2$$
where x denotes the feature vector of a pixel in the input image, $C_{1}$ denotes the background, $C_{2}$ denotes the foreground, $P\left(C_{i}\right)$ denotes the probability density function, and $p\left(x|C_{i}\right)$ denotes the posterior probability density function.

Therefore, the Bayesian classifier used to distinguish the target image will involve the determination of the prior probability function, as well as the probability density function, for distinguishing whether the pixel belongs to the foreground or background pixel. Since most of the pixels in the rectangular region should belong to the object being segmented in the determination of the initial value of the prior probability, it is sufficient to ensure that the value of $P\left(C_{2}\right)$ is greater than the value of $P\left(C_{1}\right)$. In the determination of the probability density function, assuming that they are Gaussian mixture models (GMM), the following equations are available:(2)$$p\left(x|C_{i}\right)=\overset{K}{\underset{k=1}{\Sigma }}w_{k}N\left(x|\mu_{k},\Sigma_{k}\right),i=1,2$$
where K is the number of Gaussian components in the GMM, $w_{k}$ represents the weight, $\mu_{k}$ represents the mean, and $\Sigma_{k}$ represents the Kth Gaussian component in the covariance matrix.
(3)$$N\left(x|\mu_{k},\Sigma_{k}\right)={\left(2\pi \right)}^{-\frac{d}{2}}{\left|\Sigma_{k}\right|}^{\frac{1}{2}}exp\left(-\frac{1}{2}{\left(x-\mu_{k}\right)}^{T}\Sigma_{k}^{-1}\left(x-\mu_{k}\right)\right)$$

The above formula can be used to distinguish whether a pixel belongs inside or outside rectangular box boundaries. In our method, when segmenting objects with GrabCut, the initial training data for the background GMM is the bounding pixel because the bounding box must be outside the object, so the bounding pixel must belong to the background pixel, and the object occupies most of the area in the foreground box, so the median pixel in the foreground box is used as the foreground pixel to train the data. However, in order to also have the case that the median pixel does not belong to the foreground, so to reduce the error, the range pixel method is used: the middle pixel is the center of the circle, and the average pixel within the circle area of five pixel distances around the middle pixel is used as the foreground pixel. Finally, the expectation-maximizing EM algorithm is used to learn the initial parameters of the two GMMs.

For the resultant image segmented by the BC algorithm, the similarity of each pixel $z_{i}$ to the background is denoted as $p\left(z_{i}\right)$. From the previous section, we know that BC uses the GMM constructed for foreground and background to assign likelihood values to each pixel in the image, so the GMM of the background is used to estimate $p\left(z_{i}\right)$, which is finally normalized to a probability between 0 and 1. The GMM of the foreground in the GC refinement part is calculated as in the BC step, setting the threshold T, if $p\left(z_{i}\right)>T$, then it is removed as a background pixel, and vice versa as a foreground instance pixel.

### 3.3. PONet 3D Shape Reconstruction

After extracting the furniture from the complex background segmentation, the goal is to reconstruct a 3D model of the given object. The inputs to the occupancy network ONet are images and 3D coordinates, and finally the probability of occupancy of the input points inside the object is output. Finally, the mesh model is generated based on the probability. In this study, the occupancy prediction network of PONet is proposed for the case of known camera information of input image acquisition, adding camera information to the original network, combining both local feature information and global feature information, adding detail reconstruction, and probability mixing using probability estimation of double branching, and finally outputting whether the position of the point is occupied or not to complete the reconstruction work. The reconstructed network structure is shown in Figure 2.

The occupancy prediction network proposed in this study is based on camera information known to be used for the input image capture. That is, the input to the network is the image, the camera information, and the three-dimensional coordinates (points), and the output is whether the location of the point is occupied (probability). A function $p\in R^{3}$ with observation x∈χ as input, position P mapped to a real number as output, and camera information as c∈C, the expression of the function of PONet is shown in Equation (4).
(4)$$f_{\theta }:R^{3}\times \chi \times C$$

#### 3.3.1. Encoder

The encoder of the main branch uses a RestNet-fasion CNN network [26], which completes the extraction of the global feature vector and local feature mapping from the input image. The encoder uses the RestNet-18 architecture and has an additional upsampling step to generate feature mapping, which is obtained from the fully connected layer. The outputs of the second, fourth, and sixth ResNet blocks are upsampled to 112 × 112 using bilinear differences and connected to the 112 × 112 feature map from the “Conv64” layer to form a 512-dimensional feature map, resulting in a 256-dimensional local feature region. The network structure is shown in Figure 3.

#### 3.3.2. Local Feature Extractor

The local feature mapping obtained by the encoder can only roughly get the object detail information. If we want to get the fine object surface, then we also need to extract the regional feature mapping corresponding to the point position from the regional feature mapping. This paper adopts the PFP (Perceptual Feature Pooling) method proposed by Wang et al. [27]: the input obtained camera information to project the 3D points onto the 2D image plane to obtain the corresponding positions of the points on the image, then shrink the fitted area, the feature values of the points on the feature mapping by bilinear interpolation, and finally complete the work of extracting point features. The local feature extractor process diagram is shown in Figure 4.

#### 3.3.3. Decoder

The input of the decoder contains region features and point features. Firstly, the 3D points need to be scaled to 256 dimensions through the fully connected layer to obtain the point features, while the region features are obtained by feature extraction through the encoder. The decoder part receives the point feature information, and after the conditional configuration regularization (CBN), it is processed by the ReLu activation layer and the convolution layer to generate the final output of the decoder. The internal structure of the decoder is shown in Figure 5.

In the original Onet paper, conditional batch normalization using parameters *γ* and *β* was able to restore the overall shape of the object, but it lacked certain details. To overcome this difficulty, multiple local features are used to infer the conditional normalization parameters when extracting local features. ($\gamma_{1}$, $\beta_{1}$) and ($\gamma_{2}$, $\beta_{2}$) in the above figure are inferred from multiple region features, and then the two inference results are weighted and summed to determine the final parameters *γ* and *β*. The weights *a*_1_, *a*_2_, *b*_1_, *b*_2_ are inferred using the wider receiving regions in the two regions of the region features to be inferred. The output of the decoder is the point features, and the input of the inference block is the global feature vector obtained at the encoder stage and the point features at the output of the decoder. The weight parameters in the decoder are satisfied.
(5)$$\gamma =a_{1}\gamma_{1}+b_{1}\gamma_{1}$$
(6)$$\beta =a_{1}\beta_{1}+b_{1}\beta_{1}$$

#### 3.3.4. Side Branch

From Onet’s experimental results, it is clear that the ability of a single branch to learn shapes is limited and has some limitations. To better recover the 3D model of furniture based on the pictures and improve the generalization ability of the model, a side branch structure is added with reference to some advanced methods. First, Difference of Gaussians (DOG) [28] is used to process the original input image to obtain the Gaussian difference map. This process can remove the high frequency noise components in the image, and can capture more corner point information features, so as to better learn the detail information and topology of the target object. The principle of DOG is shown in Figure 6.

The original image is first converted to grayscale, and then three different Gaussian kernels are used to convolve the grayscale image to obtain blurred images with different Gaussian scales. The processed image is sent to the encoder again for extraction to obtain the side branch prediction probabilities. Convolution of the grayscale image is performed using the formula.
(7)$$\begin{array}{ll} F_{i}\left(x,y\right) & =G_{\sigma_{i}}\left(x,y\right)f\left(x,y\right)\\ & =f\left(x,y\right)\frac{1}{\sigma_{i}{}^{d}2\pi^{d/2}}\exp-\frac{x^{2}+y^{2}}{2\sigma_{i}^{2}} \end{array}$$
where $G_{\sigma_{i}}\left(x,y\right)$ denotes the standard deviation of the *i*th Gaussian kernel, d denotes the dimensionality of the output, and *f*(*x*,*y*) denotes the input grayscale map, and the Gaussian filter results under two different parameters are subtracted to obtain the final DOG map.
(8)$$\begin{array}{ll} \mathrm{Diff} & =F_{i+1}\left(x,y\right)-F_{i}\left(x,y\right)\\ & =\left(G_{\sigma_{i+1}}\left(x,y\right)-G_{\sigma_{i}}\left(x,y\right)\right)f\left(x,y\right)\\ & =\frac{1}{2\pi^{\frac{d}{2}}}\left(\frac{1}{\sigma_{i+1}^{d}}\exp\left(-\frac{r^{2}}{2\sigma_{i+1}^{2}}\right)-\frac{1}{\sigma_{i}^{d}}\exp\left(-\frac{r^{2}}{2\sigma_{i}^{2}}\right)\right)f\left(x,y\right) \end{array}$$

The images after DoG processing retain the spatial information contained in the band between the two images, where A denotes the blur radius. The encoder in the side branch is directly used in the ResNet 18 structure, and the decoder part uses a fully connected neural network with 5 ResNet blocks.

#### 3.3.5. Probability Mixer

The prediction probabilities of the two pairs of points are calculated by the main branch and the side branch, so the final prediction probabilities are obtained by fusing the prediction results based on the contributions of the main branch and the side branch. Therefore, all probabilities are integrated using a probabilistic mixer to obtain.
(9)$$p\left(x\right)=f_{\theta }\left(\phi_{1}\left(x\right)\right)\phi_{1}\left(x\right)+f_{\theta }\left(\phi_{2}\left(x\right)\right)\phi_{2}\left(x\right)$$
where $f_{\theta }\left(\phi_{1}\left(x\right)\right)\left(i=1,2\right)$ denotes the weight of that branch in the current target prediction, $\phi_{1}\left(x\right)$ denotes the prediction probability of the main branch, $\phi_{2}\left(x\right)$ denotes the prediction probability of the side branch, and $f_{\theta }$ is the learned mapping function.

#### 3.3.6. Loss Function

To learn the network parameters from the neural network $f_{\theta }\left(p,x\right)$, by sampling random points in the 3D boundary volume of the object, i.e., for the i-th sample in the training batch, sampling *K* points $p_{ij}\in R^{3},j=1,\ldots ,K$, and then evaluating the small-batch loss function at these point locations.
(10)$$\begin{array}{ll} Loss & =\frac{1}{\left|B\right|}\overset{\left|B\right|}{\underset{i=1}{\Sigma }}\overset{K}{\underset{j=1}{\Sigma }}\varsigma \left(f_{\theta }\left(p_{ij},x_{i},c_{i}\right),o_{ij}\right)\\ & +\frac{1}{\left|B\right|}\overset{\left|B\right|}{\underset{i=1}{\Sigma }}\overset{K}{\underset{j=1}{\Sigma }}\vartheta \left(f_{\omega }\left(p_{ij},x_{i},c_{i}\right),o_{ij}\right) \end{array}$$
where *B* denotes the number of small batches, ς denotes the main branch cross-entropy classification loss, ϑ denotes the side branch cross-entropy classification loss, $f_{\theta }$ denotes the main branch network parameters, while $f_{\omega }$ denotes the side branch network parameters, and $o_{ij}$ denotes the ground truth.

### 3.4. MF-AKAZE Feature-Point Alignment

The traditional akaze algorithm detects the feature points after establishing the nonlinear scale space, uses the modified local difference binary descriptor M-LDB (modified local difference binary) to describe the feature points, and finally uses the violent matching algorithm to match the feature points. In this paper, the AKAZE algorithm is improved to obtain the MF-AKAZE algorithm. In the feature description stage, the algorithm uses FREAK (fast retina keypoint) to effectively describe the feature points, uses FLANN (fast library for approving nearest neighbors) to obtain the feature points for pre matching, and finally uses the improved RANSAC (random sample consensus) to further filter the matching results.

#### 3.4.1. Feature Detection

In the detection of feature points, the AKAZE algorithm is used to find the Hessian local maxima method, which defines the comparison space as a cube with the center of 3 pixels, compares the square with the center of 3 pixels with the nearby pixels, and compares the 9 pixels at the same position in the previous scale when the maxima are the center points, and if they are still maxima at that time, then compares them with the 9 pixels inside the next scale. If the current point is the center point, then it is compared with 9 pixels in the next scale. If the current point is still an extreme point compared to the surrounding 27 pixels, then this pixel is defined as a feature point. 

#### 3.4.2. Feature Description

The traditional AKAZE algorithm to describe image features uses the M-LDB descriptor, which is a corresponding rotational transformation on the divided network cells, and the algorithm itself is computationally complex and poor in real time. In this paper, a binary FREAK descriptor with local invariant features is used to describe the detected feature points, and the structure of FREAK is referenced to the function of acquiring image perceptual domains in the human retina, and the image information is acquired by distributing regions. FREAK is used to describe a feature point denoted by F.
(11)$$F=\underset{0\leq \alpha \leq N}{\sum }2^{\alpha }T\left(P_{\alpha }\right)$$

In the above equation, $P_{\alpha }$ denotes the sensory domain pair, i.e., the location of the current sampling point; *N* is the dimension of the feature vector.

#### 3.4.3. Feature Matching

To improve the matching efficiency in the feature point matching phase, the high-dimensional nearest neighbor FLANN algorithm is used in this paper instead of the AKAZE algorithm for BF violence matching.

The FLANN algorithm efficiently and quickly obtains the search type within a known data set according to the distribution characteristics and the required space resource consumption, and to find the nearest distance points in the feature point neighborhood based on the computed Euclidean distance, which is given by the following equation.
(12)$$D\left(x,y\right)=∥x-y∥=\sqrt{\overset{d}{\underset{i=1}{\sum }}{\left(X_{i}-Y_{i}\right)}^{2}}$$

The *D* value is inversely proportional to the matching degree of feature points, where lower *D* means the closer the distance between the feature point pairs, i.e., the feature point pairs have higher matching.

#### 3.4.4. Matching Filter

RANSAC has been widely used for matching screening work, but the algorithm can only reject a few wrong matching points, and it is difficult to screen and reject for more one-to-many matching cases. In order to improve the shortcomings of RANSAC, this paper proposes an improved RANSAC algorithm for multiple false feature point matching.

If the two feature points in the image can be matched accurately, then the single-response matrix can be expressed as:(13)$$I\;\nabla \;{\mathrm{HI}}^{\prime }$$

The two corresponding feature points in the image are denoted by I and $I^{\prime }$, ∇ denotes the proportional relationship between the feature points. The ratio between multiple matching pairs should be consistent, and if there is an incorrect matching point pair, then the single-response matrix corresponding to the wrong matching point is not consistent with the single-response matrix corresponding to the correct matching. In this paper, three matching point pairs are selected to verify the single-response matrix correspondence according to this relationship, i.e., if the feature points in all selected matching points can correspond correctly, then the triangles formed by these three points have similarity and their single-response matrices remain consistent, so the three point pairs with similar triangles can be searched for among all matching point pairs for proportional verification.

The specific steps are: firstly, the feature points are obtained by the AKAZE feature detection algorithm, and then the image is characterized by the FREAK algorithm; secondly, the matching points with multiple matching points in a special rule pair are eliminated; then one of the matching points is arbitrarily selected from the eliminated matching points as L and R, and the feature points of the domain are found by using the k-d tree as $L_{i+1}$, $L_{i+2}$,and $R_{i+1}$, $R_{i+2}$, and these three points are connected to form a triangle and compared. These three points are connected to form a triangle and compared to determine whether any two interior angles are then within a certain threshold range, and satisfying a certain range proves that the two triangles are similar, and this threshold range is specified as *d*; then we have:(14)$$\left|\cos\theta_{1}-\cos\theta_{2}\right|<d$$
where $\theta_{1}$ and $\theta_{2}$ is any two interior angles of the triangle, and the value of d is taken as 0.06. By this method, all the filtering points are detected in a continuous cycle, and finally, the matching pairs with similar triangle features are saved. The at least 3 pairs of correctly matched feature points can be calculated by Equation (13) to the single-response matrix *H*. Filtering is performed by the ratio of the single-response matrix. The matching points less than the threshold ε are saved by calculating the Euclidean distance, and the distance function d is calculated as:(15)$$d\left(x,x^{\prime }\right)=∥x-Hx^{\prime }∥+∥x^{\prime }-H^{-1}X∥$$
where ‖∗‖ is the Euclidean distance between two matching feature points, and the two corresponding feature points in the image are used as *x* and *x*’. If one of the matching points $L_{n}$ and $R_{n}$, has a relationship of $d\left(L_{n},R_{n}\right)<\varepsilon$, then they are considered as a correct matching pair; otherwise, they are rejected, where ε takes the value of 1.6.

## 4. Experiment

In this section, first, the groundwork related to the experiments is presented. This includes the selection of the dataset, the determination of the evaluation metrics, and the implementation details of the experiments; secondly, a qualitative and quantitative analysis of the 3D reconstruction part as well as the feature detection part of the registration experiments is presented, and some comparisons between the proposed method and related algorithms are made, and finally, the system implementation of the furniture AR model registration is tested.

### 4.1. Setup

#### 4.1.1. Datasets

To accomplish the task of extracting clean furniture images from complex backgrounds to achieve 3D reconstruction, this study used the ShapeNet dataset used by Nully for 3D shape recovery research. ShapeNet is a large-scale dataset containing a 3D synthetic object grid with category labels. The total data sample is 43,755 and 70%, 10%, and 20% of the samples were used as learning (training), validation (verification), and testing (test) data, respectively.

#### 4.1.2. Evaluation

In the reconstruction module, according to ONet, we used volume IoU (Intersection over Union), CD (Chamfer Distance), and normal consistency (NC) as evaluation metrics.

The IoU is a measure of similarity between two grids by calculating the ratio of the sum of the volumes of the two grids to the volume of the overlapping parts of the two grids. Therefore, the higher the IoU, the higher the similarity.

CD is one of the measures of similarity between point clouds, and it is used based on the distance between the nearest points. Therefore, the lower the value, the higher the similarity. In this study, to analyze the similarity between the correct answer grid and the grid inferred through the network, 100,000 points in each grid were randomly sampled (randomsampling) to build a point cloud and calculate the Chamfer distance.

NC is a score that indicates the normal similarity between two mesh surfaces and is a measure of the high-dimensional similarity between two meshes. Therefore, a higher normal consistency implies a higher degree of similarity.

The evaluation metrics used in the feature detection part of the 3D registration process include matching rate, time complexity, and repeatability. The match rate refers to the ratio of correct matches to the total number of feature points. The time complexity includes the sum of the time spent detecting feature points in the image, obtaining feature descriptions, and matching the features. The repeatability of feature points refers to the percentage of detected features that undergo geometric transformation or light intensity transformation in the image after feature point rereading, which is calculated based on the number of overlapping partial feature points in the image, so the superiority of the algorithm shows a positive correlation with repeatability.

#### 4.1.3. Implementation Details

Presented in this paper is a 3D reconstruction model based on Python 3.6 using Pytorch 1.0.0 implemented in an NVIDIA GeForce RTX 3060 GPU. During the training process, 1024 pixels are randomly sampled on the input image to predict the occupancy of 128 points, the image encoder part of the main branch uses the structure of ResNet-18 fine-tuning, and the final grid is generated using the multiresolution isosurface extraction (MISE) proposed by Mescheder et al. extraction (MISE) method proposed by Mescheder et al. The initial resolution set for MISE is $32^{3}$ and the maximum resolution set for active complex cells is $256^{3}$, the same as in ONet. We set 64 batches and used the Adam algorithm optimizer with a learning rate of $10^{-4}$. 

### 4.2. Results and Discussion

#### 4.2.1. Instance Segmentation

In the instance segmentation experiment, sofa, table, and chair images in the 3D-future dataset [29] are selected for the experiment. The Jaccard coefficient is used to describe the similarity between sets [30]. The larger the value, the higher the similarity. In this experiment, the 3D-future dataset contains the real image and the example segmentation result image. Therefore, the algorithm in this paper can be used to segment the real image in the dataset and calculate the Jaccard similarity between it and the segmented image for quantitative analysis. The image instances of table, sofa, and chair in the dataset are divided 30 times, and the average value of the Jaccard score is calculated.

From the analysis in Table 1, it can be seen that BC can easily extract the target edge appearance features by simple contour construction, so it can better complete the instance segmentation, such as the Sofa category; however, for the object containing the hole structure part for segmentation, since the appearance features cannot be simply extracted along the target boundary, and this paper precisely overcomes the internal structure and background for segmentation, so the score is more concentrated on other data categories.

As shown in the Figure 7, comparing several segmentation extraction methods, BC algorithm can extract the general outline of the object, but the background is extracted as part of the object, while GrabCut algorithm can extract the object inside the box, but there is no way to distinguish the hole part is extracted together, while our method BCGC can effectively extract the furniture object from the background.

#### 4.2.2. 3D Reconstruction

In order to evaluate the effect of the image-based 3D reconstruction algorithm proposed in this paper, Table 2 shows the comparison effect with the reconstruction of some previous algorithms. The input single chair, bench, table, sofa, and double chair are reconstructed using ONet, DISN, and the algorithm in this paper, respectively. It can be seen that all three algorithms can reconstruct a simple single chair accurately. In the 3D reconstruction of the bench image, the reconstruction results of ONet and DISN are smoother and better than in the present algorithm. However, in the reconstruction effect of the table, ONet lacks the fine part of the table bottom beam reconstruction and is missing. While DISN barely reconstructs the table bottom structure, the algorithm in this paper captures the structural details and can effectively reconstruct the table bottom structure. In the reconstruction of sofa pictures, ONet mesh reconstruction ignores details such as hug pillow reconstruction, while DISN can detect hug pillow information, but the effect of not reconstructing is not very obvious, and the algorithm in this paper retains the detailed information. Finally, the experiment on the double chair found that ONet will mesh complement directly for the reconstruction of the structure of objects containing holes, while DISN directly lost the backrest position information, this paper’s algorithm not only captures the backrest information, but also retains the gap existing between the two backrests, which is closer to the structure of furniture in real pictures.

In this paper, a quantitative evaluation is also performed based on the evaluation metrics provided in the previous section, as shown in Table 3. It can be seen that for the IoU results, most of the furniture categories in the ShapeNet dataset are better than the previous method in terms of the Chamfer Distance metric, but only the sofa category is slightly better in terms of the Normal Consistency metric, while all other categories of furniture models are not as good as the previous method. The reason is that the NC index is for the normal similarity between two mesh surfaces, while PONet is for detail reconstruction, which essentially solves the resolution problem on the basis of point cloud data, so the reconstruction effect for the mesh itself is not as high as the evaluation index of the algorithm directly used for mesh reconstruction. In Table 3, the symbol “**↑**” indicates that the value is positively proportional to the effect, and the symbol “**↓**” indicates that the value is inversely proportional to the effect. Bold indicates the best result in the comparison experiment.

#### 4.2.3. Feature Point Detection and Matching

In order to test the matching rate and time consumption of feature detection in the 3D registration step, this paper conducted a comparison experiment with the classical SURF, AKAZE algorithms, and the improved orb [32]. In order to verify the experimental effect of the improved feature detection and matching part of the 3D registration, a sofa image with two views from the sofa dataset in PASCAL 3D+_release1.1 is selected for the experiments. In the experimental part of verifying the repeatability effect, five sets of sofa images were selected from the dataset, and each set of images was subjected to scale transformation, viewpoint transformation, rotation transformation, and image compression transformation as the experimental dataset for verification. In the feature matching stage, the value of the error point log N was set to 8, the cosine value threshold was 0.05, and the symmetric transfer matrix error value was 1.9. The experimental results are shown in Figure 8 and Table 4.

SURF detects more feature points, but the matching pairs have fewer points, so the algorithm structure is simple and less time consuming; AKAZE algorithm has some advantages over SURF algorithm by the FED solution of nonlinear diffusion equation, so it takes less time; literature [32] takes into account the need to filter out the wrong matching points, so it takes more time compared with the previous algorithms; although the matching rate in this paper, although the matching rate of this paper can be improved, it takes more time to judge the similar triangles in the feature filtering stage, so the matching time is longer and the algorithm proposed in this paper has some defects in terms of time consumption. However, the time-consuming error is within 200 ms, which accounts for less time in the whole system, so the shortage of time consuming can be accepted.

In the experimental setup, five transformations have been applied to each group of images, so that six images are included in each group. In order to further verify the repeatability of the algorithm, the repeatability of each group of images is compared and verified from the conditions of scale transformation, viewpoint transformation, rotation transformation, and image compression, respectively. The first image of each group of images are used as the basic image, and the remaining five images are transformed with SURF, AKAZE, literature [32], and the algorithm proposed in this paper to conduct the feature point repetition rate comparison experiments. The results are shown in Figure 9.

In the scale transformation comparison, the algorithm proposed in this paper is similar to the AKAZE algorithm, but much better than the other three algorithms; in the perspective transformation and compression transformation experiments, several algorithms are less effective in the sixth group of comparison experiments, but the algorithm in this paper is more advantageous; in the rotation transformation comparison experiments, the repeatability of the algorithm in this paper is lower than that of the AKAZE algorithm, and is similar to that of the SURF algorithms. From the overall experiments, the algorithm in this paper has advantages in various indexes.

#### 4.2.4. Overall System Test Tesults

In order to verify the AR effect of the system to achieve 3D reconstruction and registration based on furniture pictures in real scenes, this experiment selects furniture sales pictures from shopping websites in the categories of single wooden stool, double sofa, wooden table, and chair, and the final reconstruction effect is shown in Table 5.

The experimental analysis shows that the ONet network cannot reconstruct furniture in real scenes with complex backgrounds, and the BCGC preprocessing algorithm proposed in this paper can make the ONet network reconstruct furniture models in clean backgrounds, but lack of details and easy to lose furniture features. After BCGC preprocessing, the proposed PONet algorithm can finely reconstruct the 3D model of furniture in the image, which maximizes the restoration of the detailed features contained in the original image, and the realistic 3D model of furniture provides a base model for AR interaction in the home environment.

The registration of furniture models into real scenes requires the use of virtual-real fusion techniques [33], and this paper only improves the scene matching algorithm for 3D registration. The final registration of the reconstructed model into the real scene to realize the AR furniture visualization system is shown in Figure 10.

## 5. Conclusions

In this paper, we propose a system for extracting furniture instances from real interior pictures and reconstructing them in 3D. Our goal is to first obtain clean furniture images using the BCGC instance segmentation method, and then perform 3D reconstruction based on 2D images. The reconstruction network model PONet is based on the occupancy network (ONet) by superimposing local feature information as well as global feature information and adding side branch probability estimation, and finally improve it for the traditional AKAZE algorithm to obtain MF-AKAZE to achieve accurate feature detection in the 3D registration process in order to enable better AR interaction of furniture models. The results show that the system is not only able to extract furniture images accurately, but also can perform better in several validation metrics compared to the Onet network, add more details to the reconstruction by analyzing the visualization results, improve the effect in terms of matching rate and repeatability in 3D registration compared to several advanced methods, and finally experimentally validate the overall functionality of the system. In this paper, the core algorithm is analyzed and improved in the experiment of 3D reconstruction of furniture pictures and realization of AR viewing of models, which has strong significance for the future efficient and real-time application of AR to business.

## Figures and Tables

**Figure 1 sensors-22-04020-f001:**
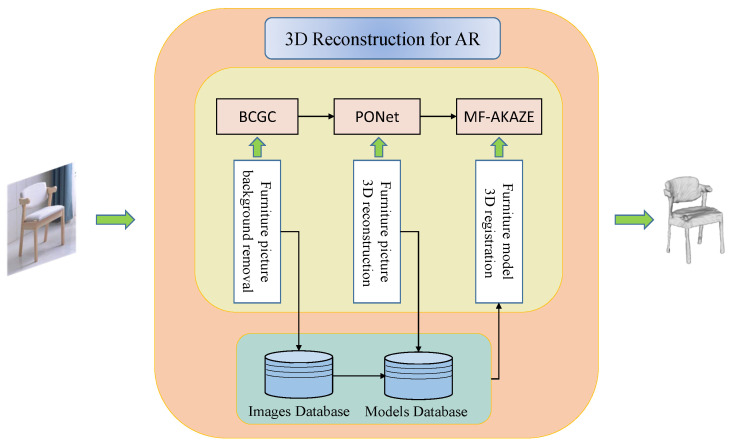
System architecture.

**Figure 2 sensors-22-04020-f002:**
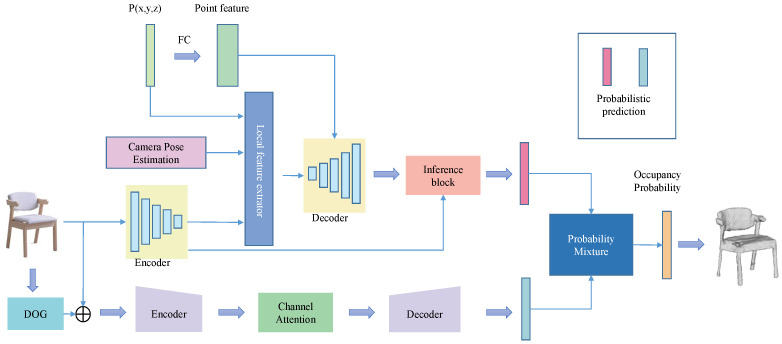
3D reconstruction of the network structure.

**Figure 3 sensors-22-04020-f003:**
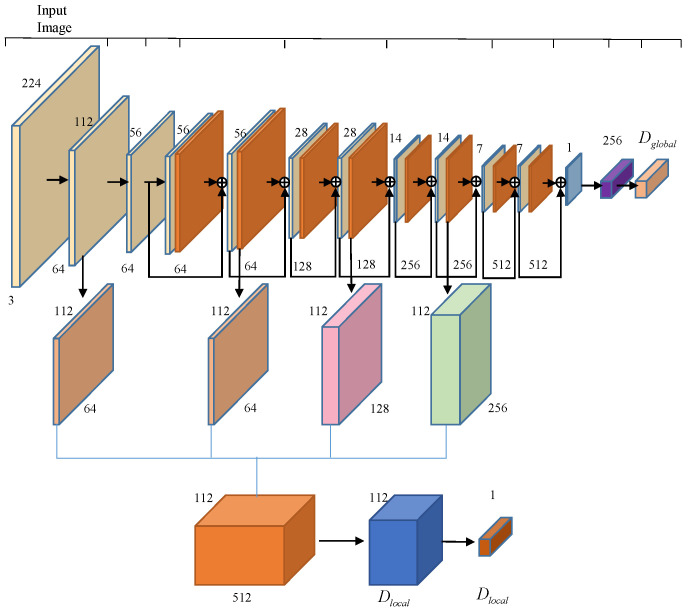
Encoder network architecture.

**Figure 4 sensors-22-04020-f004:**
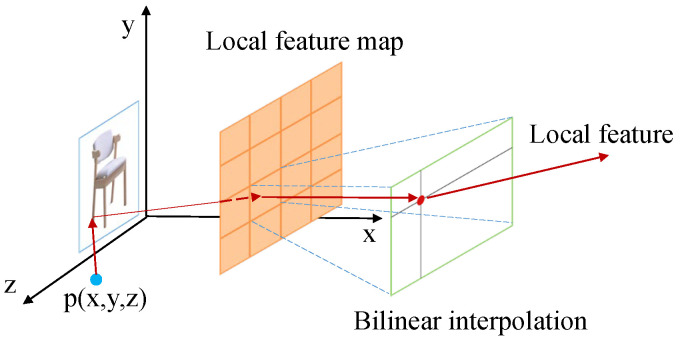
Local feature extraction.

**Figure 5 sensors-22-04020-f005:**
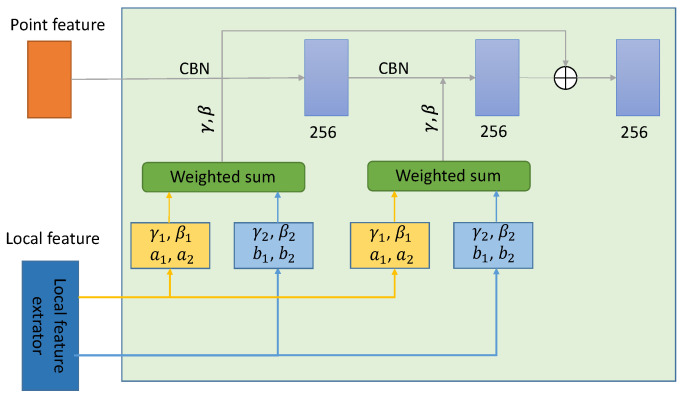
Decoder network architecture.

**Figure 6 sensors-22-04020-f006:**
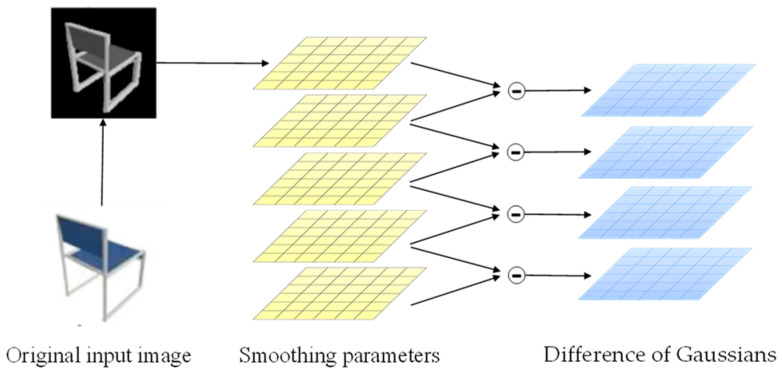
DOG processing of input picture.

**Figure 7 sensors-22-04020-f007:**
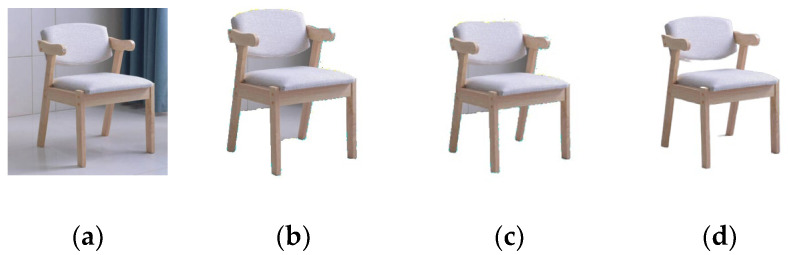
Comparison of different extraction methods. (**a**) Original image (**b**) BC (**c**) GrabCut (**d**) Ours.

**Figure 8 sensors-22-04020-f008:**
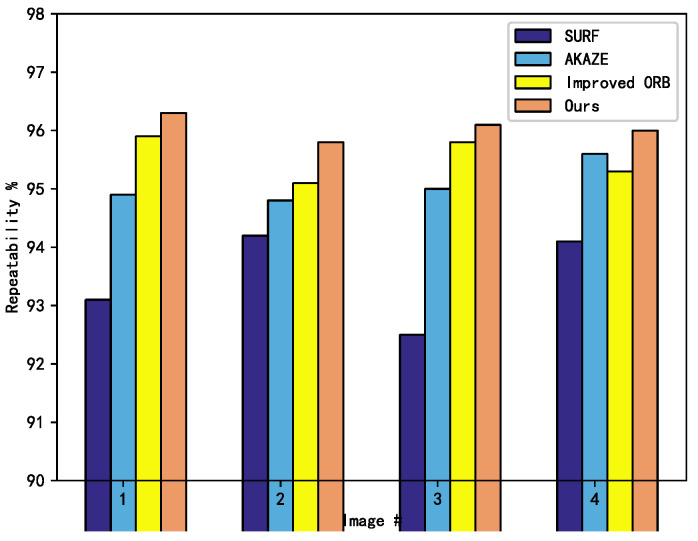
Comparison of Matching rate.

**Figure 9 sensors-22-04020-f009:**
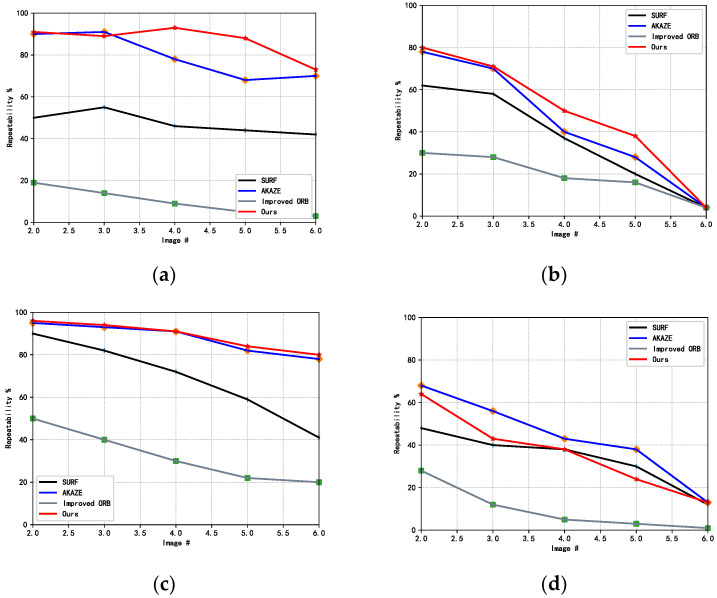
Image repeatability comparison. (**a**) Increasing Blur (**b**) Viewpoint (**c**) Compression (**d**) Rotation.

**Figure 10 sensors-22-04020-f010:**
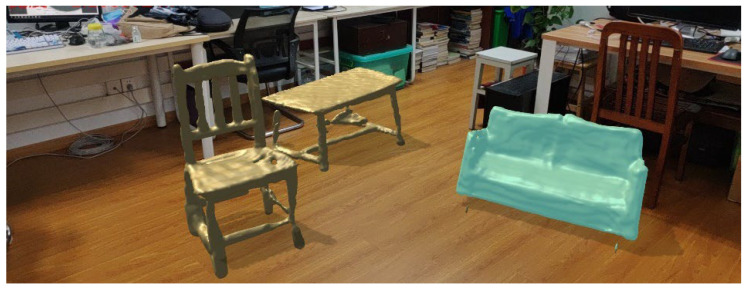
Reconstruction model AR registration results.

**Table 1 sensors-22-04020-t001:** Average Jaccard scores for different methods.

Methods	Sofa	Table	Chair
BC	97.9	85.3	80.5
BCGC(Ours)	96.8	88.7	83.1

**Table 2 sensors-22-04020-t002:** Comparison of different algorithms for furniture reconstruction.

Images	ONet [11]	DISN [31]	Ours
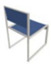	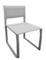	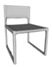	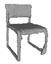
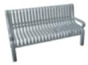	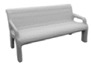	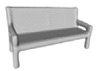	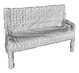
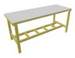	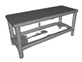	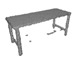	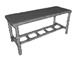
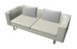	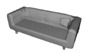	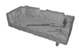	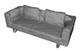
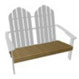	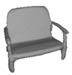	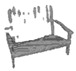	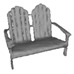

**Table 3 sensors-22-04020-t003:** Comparison of reconstruction indicators.

	IoU ↑			Chamfer Distance ↓			Normal Consistency ↑		
Category	ONet	Disn	Ours	ONet	Disn	Ours	ONet	Disn	Ours
bench	0.485	0.451	**0.509**	**0.155**	0.165	0.161	**0.813**	0.797	0.797
chair	0.501	**0.532**	0.514	0.228	0.221	**0.219**	0.823	**0.826**	0.825
lamp	0.371	0.445	**0.503**	0.479	0.268	**0.265**	0.731	**0.757**	0.729
sofa	0.680	0.690	**0.703**	0.194	0.186	**0.185**	0.863	0.870	**0.871**
table	0.506	0.538	**0.545**	0.189	**0.168**	0.175	**0.858**	**0.858**	0.851
mean	0.509	0.531	**0.555**	0.249	0.202	**0.201**	0.818	**0.822**	0.815

**Table 4 sensors-22-04020-t004:** Comparison of time consumption and average matching rate.

Methods	Detection Time (ms)	Matching Time (ms)	Total Time (ms)	Average Matching Rate (%)
SURF	301.62	97.16	398.78	92.475
AKAZE	186.33	194.41	380.74	94.075
Improved ORB	260.51	280.75	541.26	94.525
MF-AKAZE(Ours)	254.39	319.53	573.92	96.05

**Table 5 sensors-22-04020-t005:** Comparison of 3D reconstruction effect including complex furniture background.

Input Image	ONet	ONet+BCGC	PONet+BCGC (Ours)
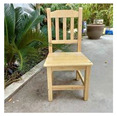	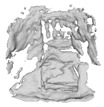	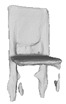	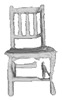
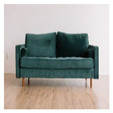	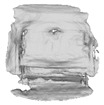	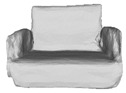	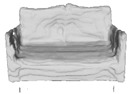
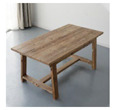	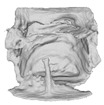	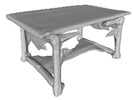	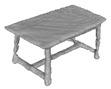
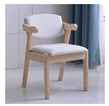	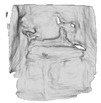	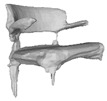	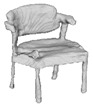

## Data Availability

Not applicable.

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
