# Peer review of "Home Environment Augmented Reality System Based on 3D Reconstruction of a Single Furniture Picture"

_sensors, 2022, doi:10.3390/s22114020_

Round 1

Reviewer 1 Report

In this paper, a system is proposed for extracting furniture instances from real interior pictures and reconstructing them in 3D.

1) This paper proposes a bayesian image background segmentation algorithm. The median pixel in the foreground box is used as the foreground pixel to train the data. But if the median pixel belongs to the background, how to reduce the error of the training. And more experiments should be compared between GC and BCGC.

2) In table 3, only part results of the category in ShapeNet are shown. Why not? I think complete results of the category should be reported. And in terms of Normal consistency, why the proposed method is worst than the other two methods. 

3)  In figure 8(a), what is the method[49]?

4) Figure 8(b) shows that the proposed method spends more time than other methods. So how to track in real-time applications?

Author Response

    1. The article does cause a large error in determining the middle pixel as the foreground pixel. Therefore, in order to reduce the training error, we will draw a circle with a radius of 5 pixels at the center of the middle pixel, and finally the average pixel in this circle region will be the foreground pixel. To increase the confidence level of several algorithms of GC and BCGC, we performed a quantitative analysis on the relevant datasets as shown in Table 2, and finally proved that our method outperforms the other methods.
    2. In Table 3, we show only some of the results on the dataset because our study was conducted for the interior furniture category, so we only performed comparative experiments on the metrics related to the furniture category. The reason why our proposed method evaluates worse metrics on NC than other methods is that our method reduces the surface normal relationship between the two grids in order to capture more details in the image for reconstruction.
    3. This is a labeling error. It has now been modified to the improved orb method proposed in literature [32].
    4. This paper only improves the method for feature detection and feature matching in 3D registration algorithm. In order to achieve the effect of 3D registration, in addition to feature matching, we also need to realize the technology of virtual reality fusion, that is, camera bit pose estimation. This paper does not mention it and directly draws lessons from other work, so we only compare the time spent in the matching stage. As an important part of feature tracking, feature detection and matching can realize real-time tracking of virtual model in real environment only by combining with virtual reality fusion technology.

Reviewer 2 Report

Very interesting research. It seems to me that there will be more applications of the results of this research in various fields. For example, for the digitisation of historical objects. I only think that the authors should refer in the related work part to the works of Autonomous Vision Group from the University of Tubingen and Max Planck ETH Center for Learing Systems. I suggest the article: "Convolutional Occupancy Networks", Songyou Peng, Michael Niemeyer. 

Author Response

Thanks to the expert's recognition of my work, I followed your suggestion and introduced the article "Convolutional Occupancy Networks" in my related work.

Reviewer 3 Report

Thank you for the opportunity to review this article.

All in all, I find the content interesting, although it does not significantly shift the quality of the output compared to other approaches. The methods used are quite well described, and the results are also presented quite clearly. 

I have a minor criticism of the presentation of the AR results. Figure 10 (two right columns) shows that the real furniture against which the 3D reconstructed content is aligned is not visible. Thus, it is unclear (1) whether there is a superposition of real and reconstructed furniture in the figures or (2) only a visualization of the reconstructed furniture in an empty room. If the latter is the case, then the paper should have better described that the 3D reconstruction section is not directly related to AR visualization and that the AKAZE enhancement is only used to improve the tracking of the environment, not to the registration of the furniture in a camera. This should be done by editing the abstract, introduction, related work, results, and conclusion.

The AR-related work section primarily focuses on the design of furniture placement in the room, but the paper's content does not address this. I would recommend expanding this section with more relevant articles to the content of section 3.4.

While I have stated that I do not feel qualified to judge the quality of the English language, I think a thorough review is needed. Expressions are often repeated in the same sentence. The sentences are too long and complicated; it is difficult to navigate their content. I have highlighted some of these sections in the attached PDF. In some cases, wrong translations of words are used (e.g., vehicle for interior decoration).

Author Response

  1. We have replaced the previous Figure 10. The reconstructed furniture is visualized by placing the reconstructed furniture model in an empty room instead of superimposing it on the real furniture. AKAZE enhancements are used only to improve the tracking of the environment, not for the registration of furniture. We have modified the last paragraph in the related work section.
  2. This paper focuses on research work in furniture image reconstruction as well as for 3D registration in which improvements are proposed for improving the efficiency of environment matching. The AR-related work section has now been modified to describe the focus on feature detection and matching.
  3. Some grammatical errors in the article have been corrected with reference to the comments you gave.

Round 2

Reviewer 1 Report

The paper has been revised according to my suggestions. And I think it can be accepted.